# A Sustainable Approach to Delivering Programmable Peer-to-Peer Offline Payments

**DOI:** 10.3390/s23031336

**Published:** 2023-01-25

**Authors:** Luca Mainetti, Matteo Aprile, Emanuele Mele, Roberto Vergallo

**Affiliations:** Department of Innovation Engineering, University of Salento, 73100 Lecce, Italy

**Keywords:** cash-like, double-spending, mobile, trusted execution environment, P2P, security, reflective architecture, one-time programs

## Abstract

Payment apps and digital wallets are powerful tools used to exchange e-money via the internet. However, with the progressive disappearance of cash, there is a need for the digital equivalent of physical banknotes to guarantee the same level of anonymity of private payments. Few efforts to solve the double-spending problem exist in P2P payments (i.e., in avoiding the possibility of a payer retaining copies of digital coins in absence of a trusted third party (TTP)), and further research efforts are needed to explore options to preserve the privacy of payments, as per the mandates of numerous central bank digital currency (CBDC) exploratory initiatives, such as the digital euro. Moreover, generic programmability requirements and energetic impacts should be considered. In this paper, we present a sustainable offline P2P payment scheme to face the double-spending problem by means of a one-time program (OTP) approach. The approach consists of wiping the business logic out of a client’s app and allowing financial intermediaries to inject a certified payment code into the user’s device, which will execute (asynchronously and offline) at the time of payment. To do so, we wrap each coin in a program at the time of withdrawal. Then the program exploits the trusted execution environment (TEE) of modern smartphones to transfer itself from the payer to the payee via a direct IoT link. To confirm the validity of the approach, we performed qualitative and quantitative evaluations, specifically focusing on the energetic sustainability of the proposed scheme. Results show that our payment scheme is energetically sustainable as the current absorbed for sending one coin is, at most, ~1.8 mAh on an Apple smartphone. We advance the state-of-the-art because the scheme meets the programmability, anonymity, and sustainability requirements (at the same time).

## 1. Introduction

The world is changing at a rapid pace, and so are payment systems. The increase of decentralized payment systems, such as blockchain [1], as a response to the big recession of 2007–2009, laid the foundation for new models of decentralized finance (De-Fi) where monetary policies are regulated by algorithms instead of authorities, and security is mathematically proven (thanks to crypto techniques instead of enforcement from supervisory bodies).

However, we would be wrong if we said that blockchain architectures are merely secure ports of traditional payment-use cases regarding distributed architectures. The novelty brought about by smart contracts and non-fungible tokens (NFTs) [2], for example, has been disruptive, and it is proof that when innovation is freed from the bridle of authority, there are advantages (even for the regulators). In particular, the programmability feature brought about by smart contracts makes it possible to program cash flow and, thus, integrate delivery processes and payment transactions into one system. Even fiat money is becoming crypto; central banks and governments are working toward creating digital equivalents to traditional money. The set up of central bank digital currency (CBDC) [3,4] exploratory initiatives [5,6], such as BCE’s digital euro (D€) [7,8], aims at delivering the same exciting applications of the most modern cryptocurrencies while being under the umbrella of a risk-free environment brought about by regulatory bodies. As reported in [9], in mid-July 2020, at least 36 central banks have published CBDCs. The industry and the scientific community are striving to find alternatives when implementing CBDCs [10,11,12].

Among the expectations of CBDCs, there is the possibility of having cash-like features [13], such as offline payments. The reason behind the requirement for cash-like features is anonymity—a fundamental human right [14] that not even blockchains can ensure, despite some very interesting works that exist in the literature [15,16,17]. We are always connected to the internet and we leave digital traces everywhere. Moreover, regarding blockchain information, our payments are publicly available and hidden identities could be reverse-engineered. As a response, today we are witnessing a new breed of payment systems being developed in addition to digital money, including peer-to-peer (P2P) networking to enable offline payments.

The problem behind offline payments is the double-spending attack. One of the reasons that banks exist is because of their role in mediating transactions: financial intermediaries are here because we have to be certain that those who spend digital money will not retain copies in their wallets. In the absence of a trusted third party (TTP), blockchains solve the double-spending problem by means of consensus algorithms supported by proof of work (PoW) or proof of stake (PoS) techniques, but in offline mode, peers cannot count on public validation for privacy reasons. Offline schemes built on top of the Bitcoin blockchain, such as the lightning network [18], are still under examination due to concerns about their actual privacy levels [19].

Other attempts to solve the problem exist in the scientific literature. For example, the authors of [20] noted that the double-spending problem is a serious issue that is not mathematically solved for on-chain payments, due to the possible 51% attack. In [21], the authors used secure element-specialized hardware provided by payment cards and a NFC chipset (of which, modern smartphones are equipped) to protect merchants from the double-spending attack in offline payments. Although its focus is on protecting the merchant’s counterpart, unfortunately, programmatic access to secure element features is constrained by some smartphone vendors (e.g., Apple). Moreover, [22] focused on shops and resolved the problem of e-cash anonymity control without TTP. Nevertheless, the authors have left the double-spending and double-deposit checks to the bank, i.e., when the shop transferred a coin to the bank and the bank deposited it to the bank account, hence, the mechanism was not tempestive as it could not recognize and stop the double-spending attack when it happened. In an old work [23], the authors stated that anonymity should be treated as a control parameter facilitating the flexibility of the level of privacy of note holders. Consequently, the double-spending problem was not even mentioned. At the time of writing (1997), digital payments were not so widespread and diverse, and the need for a fully anonymous and secure payment scheme was not as urgent as it is nowadays. In [24], the authors focused on a slightly different aspect: they solved the problem of double-spending without TTP using an ACJT group signature scheme [25], but transactions were still online. Moreover, tests using traditional blockchain consensus algorithms have been performed on mobile and wearable devices [26]. Results show that existing methods based on PoW have tremendous negative impacts on battery life, besides the obvious need to remain connected to the internet. In [27], the authors achieved double-spend tracing without TTP using the leakage of knowledge. Nevertheless, there is no way to block the transfer of e-coins in the act of double-spending. Finally, to prevent the double-spending of a coin, in [28], the merchant verified (online with the bank) that the coin was still valid, so the identity of the payee was revealed.

As it is evident from this analysis of the existing research works, the problem of double-spending is only partially solved. In particular, existing solutions do not block the double-spending attack when it happens. Instead, they use some tracing mechanism that, when needed (e.g., in case of problems or during a check), can help one know if that e-coin was double-spent, find responsibilities, and recover previously safe conditions. So an existing research challenge is to provide an offline P2P payment scheme that makes it impossible to double spend [29]. To make it even more challenging, as per the requirements of different CBDC exploratory efforts, the e-coin should be programmable, meaning that the e-coin should be able to execute some additional logic when it is spent. Finally, the scheme should be sustainable, as complex cryptographic algorithms (such as the ones used in the considered related works) could quickly drain the batteries of mobile devices.

In this paper, we propose a P2P offline payment scheme that meets four main requirements at the same time, i.e., (i) it makes it impossible to double spend e-coins; (ii) privacy and anonymity are always 100% guaranteed; (iii) it allows attaching logic to the e-coin in order to make it programmable; (iv) it consumes a reasonably low quantity of currents to spend single e-coins, in the sense that a full battery charge should be enough to spend a few hundred currency units on a common smartphone. The idea presented in this paper is to encapsulate each monetary unit in a one-time program (OTP) [30] that will exploit the trusted execution environments (TEEs) of modern smartphones and transfer itself from the payer to the payee via a direct IoT link. A OTP is an open issue itself; the idea is to use the reflection architectural pattern [31] to wipe the business logic out of the clients and allow the financial intermediaries to inject certified payment codes into the payer’s device, which will execute asynchronously and offline at the time of payment.

As proof of the feasibility of the scheme, we developed a prototype for iOS devices. On iOS, the TEE is called a secure enclave [32] (SE). The simplified scenario is: (1) the customer asks to withdraw D€s from his/her mobile wallet; (2) the bank wraps and signs every 1 D€ in an OTP and sends it to the customer; (3) OTPs are exchanged across citizens for anonymous payment, even offline; (4) any citizen or merchant can deposit the OTP to his/her bank at any time. Notice that no synchronization between wallets and banks is needed at any time (it would lead to privacy issues). Moreover, similar to cash, a bank account is not needed to have currency in one’s own mobile wallet. The current absorbed for sending 1 D€ is, at most, ~1.8 mAh on an iPhone 7.

The rest of the paper is structured as follows. Section 2 describes the proposed method, with a preliminary focus on OTPs and reflective architectures. In Section 3, we provide an assessment of energy consumption on a real device and provide a security evaluation. In Section 4, we discuss the main findings of our work. Section 5 summarizes the conclusions and future research efforts.

### Main Contributions

In this paper, we advance the cybersecurity and privacy fields by presenting a novel approach to executing OTPs on mobile devices, particularly for mobile payments. OTPs were conceived in 2008 as a solution for the double-spending problem; today, they remain a theoretical conceptualization of something that, in practice, cannot be implemented because of hardware limitations on commercially available devices. A feasible and practical solution is presented in this paper, by leveraging the reflective architectural pattern. This approach preserves user privacy and guarantees security against the double-spending issues; moreover, it satisfies the programmability requirements of digital currencies, as per the mandates of several CBDC initiatives. In contrast to the pure cryptographic approaches, it is based on an application-level protocol; hence, it gives the flexibility of wrapping each monetary unit in a specific program that can be augmented with additional logic besides the one needed to transfer the coin itself. Finally, energy consumption is not low but it fits well with common usage scenarios for cash-based daily transactions.

## 2. Materials and Methods

### 2.1. One-Time Programs

A OTP is a computational paradigm that was defined in 2008 and is geared toward security applications. OTPs can be executed on a single input, specified at the runtime, and can be akin to a black box function that may be evaluated once and then “self destructs.” The OTP paradigm serves many purposes, e.g., software protection, temporary transfer of cryptographic ability, etc. In particular, OTPs lead to electronic cash or token schemes, where coins are generated by a program that can only be run once, avoiding the double-spending problem. Since its first definition, the OTP paradigm has received attention from the scientific community due to the evident difficulty in creating concrete implementations. It is impossible to avoid any piece of software to be copied and run again without using additional instrumentation. In the literature, different approaches for OTP implementation exist, from using dedicated hardware and complex cryptographic techniques (as in the first paper itself [30] and [33]) to extremely compelling approaches based on quantum computing [34,35,36]). In particular, Ref. [33] seems to be a very promising idea because it uses hardware that is already present on most modern mobile devices.

### 2.2. Reflective Architectures

The reflection architectural pattern [31] is a mechanism aimed at dynamically changing both the structures and the behaviors of software systems, meaning that fundamental aspects of software-like structures and function calls can be modified at runtime. An application implementing the reflection pattern is split into two parts:The meta-level. It includes information about the property of the selected system, making the software self-aware. When the information kept by the meta-level changes, the subsequent base-level behavior is affected.The base level. It includes the concrete application logic, and its implementation builds on the meta-level.

The meta level gives the software a self-representation of its own structure and behavior, and consists of so-called metaobjects, whose aim is to represent and encapsulate information related to the software (e.g., type structures, algorithms, or even function call mechanisms). Every metaobject encapsulates information about a specific aspect of the structure, behavior, or state of the base level. Nevertheless, not every system detail is adaptable to be encapsulated in a metaobject. In particular, the encapsulation process should affect only details that are likely to change or that vary from customer to customer.

The base level defines the application logic based on the implementation of the metaobject, so that it remains independent from the aspects that are likely to change. For example, a base-level component may communicate with another one only if there is a metaobject implementing a specific user-defined function call mechanism. This gives extreme flexibility to software maintenance, because when the metaobject changes the way the base-level components communicate, the base-level code is not affected. Information and services provided by metaobjects (such as location information about the components and function call mechanisms) are used by the base level, allowing it to remain flexible and keep its code independent from aspects that can vary in time. Using the metaobject’s services is very useful because base-level components do not need to hard-code information, they only have to consult appropriate metaobjects for this information.

### 2.3. Hardware and OS Setup

All tests were carried out on a 2016 iPhone 7 equipped with a 3000 mAh battery, the operating system on the smartphone was iOS 16.0; the phone used was free of accessory applications (except built-in apps), Wi-Fi was off while Bluetooth was on; the difference in the use of one over another was almost zero. The choice of developing a prototype on an Apple device was aimed at proving the feasibility of our payment scheme in the “worst case”, i.e., the most constrained mobile environment present on the market today.

### 2.4. Personal Area Networks (PAN)

The connection between two mobile devices must be safe and reliable at the same time. Technologies, such as Bluetooth low energy (BLE) and Wi-Fi, allow these features. In order to have a fluid and immediate system, the use of multipeer connectivity (and, therefore, P2P) is the most valid option because it does not require many operations by the user to pair between two devices.

In this work, we consider Apple iOS as a target testbed platform. Apple’s multipeer connectivity framework [37] facilitates finding adjacent device services and corresponding with them via message-based data, streaming data, and resources (such as files). For the underlying transport in iOS, the framework employs Bluetooth personal area networks, peer-to-peer Wi-Fi networks, and infrastructure Wi-Fi networks.

### 2.5. Secure Enclave Keychain

We took inspiration from [33] by strongly taking into account the many advantages of the TEE, which is called the secure enclave (SE) [32] on Apple devices. To ensure maximum information security, the use of dedicated hardware components is an ideal solution for our purposes. The ability to generate biometric keys through SE enables secure storage in the keychain. SE is a dedicated secure subsystem integrated into Apple’s system-on-a-chip (SoC), designed to protect critical user data even if the application processor kernel were to become compromised. It is segregated from the main processor to add an extra layer of protection. Its design is very similar to that of SoC and includes a boot ROM that creates a hardware trust root, an AES engine for quick and secure cryptographic operations, and protected memory (see Figure 1). Despite not having its own storage area, SE features a technique allowing to safely store data on a connected region (separate from the NAND flash memory) that the processor can use for programs and the operating system.

### 2.6. Diffie–Hellman Algorithm with Zero Knowledge Proof

The Diffie–Hellman (DH) algorithm is a widely used method for securely exchanging keys over a public communication channel. The Diffie–Hellman with zero-knowledge proof (ZKP) algorithm [38] extends the standard DH algorithm by adding a mechanism for proving knowledge of the shared secret key without revealing the key itself. This is useful in situations where one party wants to prove to another party that they possess a specific secret key, but they do not want to reveal the key itself to avoid man-in-the-middle (MITM) attacks. ZKP enables proving the possession of a secret key without revealing it. It is based on the idea of interacting with a verifier through a series of questions and answers where the prover can demonstrate that he knows the secret value but it is not possible to obtain it from the answers.

### 2.7. Proposed Method

In our scheme, a OTP is composed of a sequence of steps (behavioral section, BS) and a set of parameter definitions (static section, SS). This structure, which represents the meta-level in reflective architectures, orchestrates the transaction of a single digital coin, whose identification code is also embedded in the SS. Among the steps, in the BS, the OTP contains a pre-condition check (this determines if it can be executed or not), a self-transfer instruction (so it could be passed from one peer to another), and a self-destruct instruction.

The OTP is issued at runtime by the bank at the time of withdrawal. The bank withdrawal service picks a coin from the customer’s bank account, generates the BS and SS, and sends the JSON payload to the client’s app, which contains the concrete implementation of methods (base level in reflective architectures) and will run it asynchronously for spending it.

The idea is to manage a secure sandbox, which is a data structure that can encapsulate information regarding OTPs, by means of a custom Swift framework, which is a pre-compiled library that can be provided to third-party developers.

As shown in Figure 2, the sandbox (encrypted through SE) contains the OTP itself and an exchange key. The pre-condition check consists of checking if the OTP is correctly signed by a bank and if there is a non-void field containing an exchange key. When the payer wants to make a payment, the payee must send a key (the exchange key) for each OTP s/he’s asking to receive, and those keys have to be correctly issued (signed) by a bank. Exchange keys are issued by banks and sent to wallets asynchronously with respect to payments (e.g., when the wallet is running out of exchange keys, when there is a network connection, etc.). The payer’s app will save the exchange key in a logical position paired with the spent OTP, which is the sandbox itself. After the exchange, the payer will end up with a complete sandbox, and then it is no longer usable, while the payee will have a partial sandbox with only the OTP inside. The self-destruct instruction will remove the sandbox from the protected memory. If there is a missing sandbox deletion, the framework will still recognize it as non-usable, so it would not be possible to perform any kind of action with that element. This allows for preventing double-spending, which involves the reuse of the same information.

In our scheme, the OTP model contains several checks and support versions. Using the framework, the payer application is able to check the correctness of all OTPs received from the bank and send them using a secure local channel. In this context, a user who wants to withdraw one D€ from his bank account receives an encrypted version of the following JSON (Figure 3), which represents the sandbox.

Using the reflective architectural pattern, the base level of OTP is composed of the *callMethod* section; the meta-level of OTP is composed of the *data* section. Using the *schemaVersion* attribute, the application is able to recognize the version of OTP. This attribute can be useful for many purposes.

Now, suppose that Alice needs to send N D€ to Bob. Bob needs to send N exchange keys to Alice. When Alice receives the exchange keys from Bob, she is ready to send (run) the required number of OTPs.

The execution of OTP consists of different atomic steps. In particular:*checkIntegrityOfOTP*: the bank generated the OTP and inserted a control hash in the *checksum* field. The payer app generated a hash of the received list and compared it with the *checksum* field. If they are equal, the procedure continues.*checkTokenFormat*: the application checks that the *token* in the *data* field respects the expected format. If they are equal, the procedure continues.*checkCallMethod*: the application checks that the methods in the OTP’s *callMethod* field are in order and consistent with the framework. If they are equal, the procedure continues.*setupConnection*: a secure connection and channel for sending exchange keys and OTPs is established. If the connection is successful, the procedure continues.*buildSandbox*: the payer’s app builds sandboxes by entering the exchange keys. If the build is done correctly, the procedure continues.*sendOTP*: the payer sends the sandboxes with the exchange key field empty.*destroyOTP*: the payer’s app deletes the newly constructed OTPs from the secure memory.

At the end of the procedure, Alice will have a complete sandbox while Bob will have the original version of OTP that Alice received from his bank. It is interesting to note that even if the destruction of the OTP from Alice’s application is not complete, the OTP would be unusable since the KEY field is full and the complete sandbox is in the secure memory. This aspect resolves the double-spending problem. This completes the use case reported in Figure 4.

The edge-side application, which is installed on the user’s phone, will be nothing more than a player that will run the OTP containing information about the token itself and the commands needed to make auto-transfer and self-elimination. This idea eliminates the problem of double-spending while maintaining the cash-like factor, the money would be available without having any kind of traceability, both locally and remotely. Moreover, the steps listed in the BS could be enriched to add further logic and, hence, enable the programmability of tokens.

It is assumed that the developer of the wallet app, when it has to implement its application, must not directly manipulate OTPs and exchange keys. This means that the responsibility must be external to the programmer. We met this requirement by developing a framework, a fully pre-compiled swift code library so that the code would not be accessible or editable from external sources. In this way, the final developer will only have to include this element within its application, which will have total responsibility for managing the OTP life cycle, while delegating to the programmer the development of the user interface for the app (Figure 5).

As previously seen, the interaction between the payer and payee happens offline, unlike the withdrawal and deposit, which obviously use an internet connection (Figure 6). The protocol uses an asymmetric key to cipher OTPs. In this way, the customer (holding the public key in the wallet app) is certain that every received OTP is authentic (i.e., generated by authoritative banks) and that it has not been modified by attackers.

Steps 2–6 in the previous image show the exchange between the exchange key and signed OTP in P2P connections. In the communication between private users and banks (steps 1 and 6), regular secured connections are used (HTTPS). The management of the business logic wrapped in the OTP is left to the entity that distributes the coin; the cryptographic process that is performed through the SE is only related to the additional layer of security needed when the data are in the smartphone, to handle the double-spending issue. The horizontally dotted line delineates two sections: the lower section does not need a connection to the internet network, while the upper area does. Users do not need to be connected in any way to the internet while exchanging OTPs. If the merchants, or any users owning OTPs, want to deposit coins to their bank accounts, they transfer the OTPs via a deposit service exposed by their bank, which will unwrap each token from the OTP and store it in the customer’s account.

## 3. Quantitative and Qualitative Evaluation

### 3.1. Energy Consumption

We developed an application prototype and carried out some ramp tests related to an increasing amount of OTPs to transfer. The time needed to transfer an OTP includes the time of storage and secure management of information, besides the networking times. The following amounts of OTPs were sequentially sent: 5, 10, 20, 30, 40, 50, 100, 200, 300, 400, and 500 (Figure 7 and Figure 8); the receiving device was very close to the transmitter (a few inches). The connection that enabled the multipeer at this time was Bluetooth; the other usable technologies (such as Wi-Fi) were temporarily off (the performance difference using Wi-Fi instead of Bluetooth was approximately zero). For each amount of OTPs in the ramp, we subsequently performed 50 repetitions and then we calculated the mean value (i.e., 5 OTP sent 50 times, 10 OTPs sent 50 times, and so on). The interval between repetitions was zero, so everything was done sequentially.

The test may not seem that significant because one may think that nobody could execute 100+ transactions per day. Nevertheless, considering that at the current state of the protocol each transaction involves only 1 digital currency (e.g., USD 1 or EUR 1), it could be very easy for a payer to settle 100 transactions in a day. With this current limitation, it makes sense to check how many transactions could be settled with a full battery.

In the upper part of Figure 7 and Figure 8, it is possible to see the mean time needed to send a variable quantity of tokens, followed by the graph that indicates the standard deviation. For a large number of OTPs to transfer, time is a significant value, although the diagram follows a linear trend. More than 6 minutes are needed to send 500 OTPs. After a start-up phase with more irregular values (as can be seen from the standard deviation graph), the time needed to send OTPs grows linearly, taking around 0.78 s per OTP.

To investigate how the latency builds and how energy is spent, we look at the previous graph, which takes into account the time needed to send and save OTP, without the cryptographic work made by the SE (Figure 9 and Figure 10).

The trend grows approximately linearly but the time is significantly smaller. This means that the use of technologies dedicated to encryption and data security, such as the SE, needs a considerable amount of time and energy. Even the use of the battery in the case of non-use of the SE is much lower and almost negligible. In Figure 11, it is easy to observe that our scheme certainly has a consumption that should be taken into account. In this graph, on the x-axis, we have the time instead of repetitions, i.e., the time needed to perform the ramp test. Thus, after 400 s, the battery is almost drained.

We observed a certain correlation between the execution time of the tests and the temperature sensed by the sensor placed on the device battery. Figure 12 shows the temperature chart using values collected during the tests.

To identify the role of battery temperature in the increase of transfer times, we performed an additional test where we sent a fixed amount of 100 OTPs, 50 times (Figure 13). The more the temperature of the device increased due to the constant sending of OTPs, the more the needed time increased.

Sending 100 OTPs took 39 s at rest, increasing to 44 at the top of the energy expenditure. In this case, there was a 60 s rest between repetitions, but evidently, it was not enough to cool down the battery.

During the experiments, we observed a linear correlation between energy consumption and the number of OTPs sent. To compute the approximate energy needed to transfer a single OTP, we picked the highest energy consumption value from the experiment, which corresponded to 892 mAh for sending 500 OTPs. We took the highest value to consider the worst case, i.e., when the battery temperature was high and, hence, less efficient. From a simple calculation, we obtained:(1)$$E_{OTP}=0.892\;\mathrm{Ah}/500=0.001784\;\mathrm{Ah}=1.784\;\mathrm{mAh}$$

This value implies that our scheme can exchange up to 1682 OTPs with a battery capacity of 3000 mAh. Obviously, this estimation is ideal since it does not take into account the intrinsic consumption of the device, such as the OS, other network usages, and the display. However, we believe that allowing to exchange up to 1682 units of currency (e.g., D€) with a fully charged battery is a good result related to regular daily cash usage.

### 3.2. Performance Evaluation

In this section, we report some additional performance metrics, besides the time metrics we presented in the previous subsection.

In Figure 14, we show a screenshot of Apple’s Instruments app, reporting the percentage of CPU usage during a transfer of 100 OTPs. Only the 2 performance cores (CPU 4 and 5) are reported. The loads over the other 4 efficiency cores (CPU 0–3) are negligible and, hence, not reported.

In the CPU usage graphs, there is only a short heavy load over 1 of the 2 CPUs at the beginning of the payment process. The loads over the 2 CPUs remain low, most of the time at 10%, with some rarer peaks at 20%.

Moreover, as a usability performance evaluation, we report the different networking setups used by the iPhone to check which conditions our system works under. Remember that our system is aimed at replacing physical cash, a means that works in any condition, including in airplane mode. Table 1 reports all possible working configurations. The different setups do not make substantial changes in the performance of the scheme.

The only condition in which our system refuses to work on iOS is when airplane mode is on and the Wi-Fi and Bluetooth peripherals are turned off. However, Bluetooth is normally allowed on airplanes, so we can state that our scheme can work in every condition.

### 3.3. Security Evaluation

In this section, we present a qualitative evaluation of the security of the proposed solution. Our goal is to assess the resilience against potential threats and vulnerabilities and to identify any weaknesses that could be exploited by attackers. To conduct the evaluation, we employed a variety of techniques including:Vulnerability analysis: involves identifying and analyzing potential weaknesses that could be exploited by attackers to gain unauthorized access or disrupt the normal operation of the system;Input data analysis: involves evaluating the data that are input into the protocol to ensure that they are valid, accurate, and in the correct format, in order to reduce the risk of a security breach;Dependency analysis: involves identifying and evaluating the various external software libraries, frameworks, and other dependencies that the protocol relies on;Authorization analysis: involves evaluating the access controls and authorization mechanisms that are built into the protocol to ensure that they properly protect the system from unauthorized access and use;

Figure 15 represents all of the steps that an attacker would have to take to breach the system.

Through these methods, we aimed to gain a comprehensive understanding of the security and to provide recommendations for improving the security posture. This evaluation is reported in Table 2.

With regards to the first row (the exchange process), there are no issues in case of a missed connection or battery failure—for two reasons. The first involves cash-like payments; the two users are close as if they are exchanging physical coins. The possibility of a missed connection because two users walk away or voluntarily turn off the connection has not been considered unrealistic. Second, we can expect the presence of an ACK whenever a user receives an OTP. If the sender does not receive the ACK, the connection will be closed without the loss of an OTP. Suppose that the sender needs to send 3 OTPs ( OTP1, OTP2, OTP3). He sends OTP1 and receives ACK1; he sends OTP2 and does not receive ACK2. In the sender’s hand, he built the sandbox for OTP2 with the key NOT NULL and sends OTP2. If the receiver has a problem, the sender will again try to send OTP2 or close the connection without loss. The only possibility to lose an e-coin is if the sender shuts down due to a battery problem. In this case, there are two scenarios. First, the sender’s phone shuts down after the sending step. This is not a problem because the receiver has the OTP. Second, the sender’s phone shuts down during the sending step. In this case, the sender loses his e-coin because, during the connection setup, the e-coin is removed from the SE.

## 4. Discussion

In Table 3, we present a comparison between our approach and the other approaches presented in Section 1. The approach we are evaluating allows for adding extra functionality without the use of an internet connection, thus allowing our scheme to be completely independent from th TTP and not dependent on any other nodes but those involved in the transaction. The technology related to SE provides us with an additional level of security and privacy since, in addition to having cryptographic technology in our data, which will be processed and saved securely, it is not needed to share and transfer to people outside the transaction of the payment information, making it private and secure on the smartphone. The energy cost is non-negligible. The execution time and energy consumed are significantly higher than distributing the system over several nodes, despite being acceptable in terms of the classic usage of cash.

The only effective comparison we can make with other approaches presented in the literature is with [26], which is the only one providing energy measurements. It is based on creating a blockchain-like infrastructure using multiple nodes for the approval of the transaction. In this case, the whole procedure is managed through one or more smartphones; therefore, it is significantly faster and less energetic than our approach. This is because it divides the computational efforts of different tasks on many devices, but this makes customers lose the privacy of payments. Other approaches have other drawbacks. One is based on external hardware, which is not always available to programmers (the secure element); many schemes are not prompt in the act of detecting and preventing double-spending attacks and others require bank accounts. Finally, some schemes need an internet connection. Thus, our scheme is more similar to cash, which we added for completeness in the first column. Moreover, it is the only approach capable of adding programmability to coins, thanks to the implementation of the reflective architectural pattern.

The main difference that can be identified between the presented approach and the different blockchain approaches considered in this evaluation is related to the network connection. It is not necessary to have a connection to the internet network to carry out a transaction. This also implies a second profound difference, we do not need nodes performing computationally onerous work to carry out a money transfer; the whole process is carried out independently of the devices in the possession of the users, taking advantage of communication technologies such as BLE. So the approach used is profoundly different from the blockchain approach.

The comparison with the secure element approach [21] that we presented in this paper relies entirely on the presence of a TEE on the user’s device. Hence, a simple secure element, such as the one present in e-sims or bundled within the devices, is not suitable for the purposes of the presented protocol, as secure elements are only intended to safely store secrets, while TEEs are full subsystems able to perform crypto calculations.

Finally, we must discuss the main achievement of our scheme in terms of privacy benefits. In the path toward the progressive migration to a cashless world, online payments threaten the privacy of citizens because of the need for an internet connection, even in decentralized systems. Our scheme satisfies the privacy requirements at different levels:At the hardware level, the use of the TEE ensures high privacy protection because it can be accessed only using biometric information (or personal codes), while being physically separated from the main CPU.At the PAN level, data are exchanged only between the two involved peers, who establish a private link exempt from MITM attacks.At the internet level, no connection is needed when spending e-coins, nor is it needed to synchronize with servers at any time.At the application level, OTP issuers (banks) only know when customers withdraw and deposit e-coins. OTPs are anonymous and are not bound to customers. No one, except the owner, can know how e-coins are spent.

## 5. Conclusions

In this paper, we presented a novel P2P offline payment scheme that satisfies the requirement of privacy for the digital equivalent of cash. As required by many CDBS exploratory initiatives, such as the digital euro, the payment means of the future shall allow citizens to perform cash-like payments, i.e.,

Digital currency will be minted only by a TTP-centralized system.An internet connection should not be mandatory to perform a P2P transaction, even a posteriori (e.g., no late synchronization with banks or other TTPs).Privacy and anonymity in payments.Digital currency cannot be cloned, avoiding double-spending.A bank account is not required to retain digital currency in the wallet.Energetic sustainability.

As an add-on, the programmability feature is a strong further requirement, inspired by the blockchain’s smart contracts. Based on the evaluation we performed, our approach is the only one to meet all requirements listed above at the same time. Our approach consists of wrapping each monetary unit in a one-time program (OTP) at the time of withdrawal (i.e., when the e-cash is transferred from the bank account to the wallet app on a mobile). At a high level, the OTP contains a pre-condition to check whether it was already run or not, the instruction to transfer itself from the payer to the payee via an IoT link, and the self-destruction instruction. The payee is not obliged to have a bank account, as s/he could spend the OTP with other payees, and so on, until a payee wants to deposit the OTP in his/her bank account. At this point, the coin is unwrapped and deposited online. The OTP paradigm is an open research issue itself, we solved the issue by means of the reflective architecture pattern: the OTP consists of a certified list of meta-instructions, so the business logic is wiped out from the wallet app, as it contains only the concrete implementation of atomic functions. The wallet developer will only have to include a pre-compiled software library within its application, which will have total responsibility for managing the OTP’s life cycle, while delegating to the programmer the development of the user interface for the app. The scheme uses the trusted execution environment (TTE) of a modern mobile CPU, taking inspiration from [33].

We tested our novel payment scheme on an Apple iPhone. We chose the Apple platform as a target testbed because it is the most constrained environment for the smartphone market, both for hardware features and available SDKs. Results confirm the energetic sustainability of the approach, as current absorption is at most ~1.8 mAh when spending a single coin.

A limitation of our scheme is the need to wrap every single coin in a specific OTP, so if a payer wants to spend N coins, N OTP executions are needed, so the energy and time spent increase proportionally. One logical solution is to take inspiration from physical banknotes and create OTPs with different denominations (e.g., 5, 10, 20, 50, 100, 200, and 500). The problem in this case is that the scheme shall be modified in order to include the handling of the change, which would tremendously complex the payment process. So further research efforts are needed to check the pros and cons of multiple denomination OTPs.

Another limitation is the high risk of injecting meta-codes into user devices. The library embedded in wallet apps checks for a valid signature before executing the code, but a further cybersecurity assessment involving, for example, some penetration tests, should be envisioned. A related risk is that a malicious wallet app could trick the embedded library and modify or totally halt the execution of OTPs. Further research efforts are, hence, needed to clarify and possibly solve the risks we found.

## Figures and Tables

**Figure 1 sensors-23-01336-f001:**
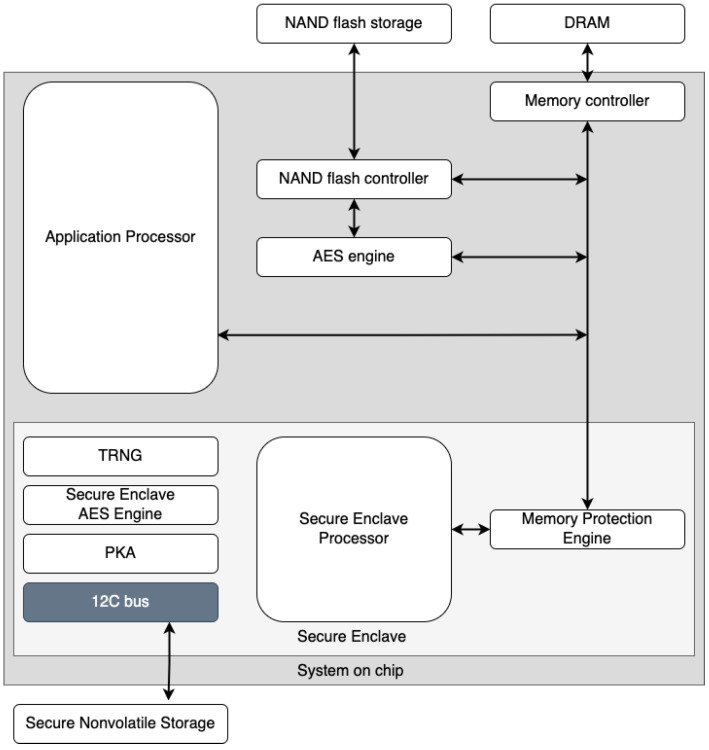
Hardware–software interaction of the Secure Enclave.

**Figure 2 sensors-23-01336-f002:**
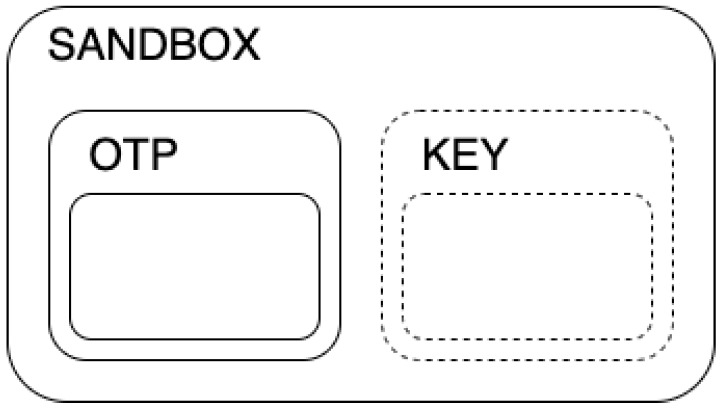
Structure of an OTP sandbox.

**Figure 3 sensors-23-01336-f003:**
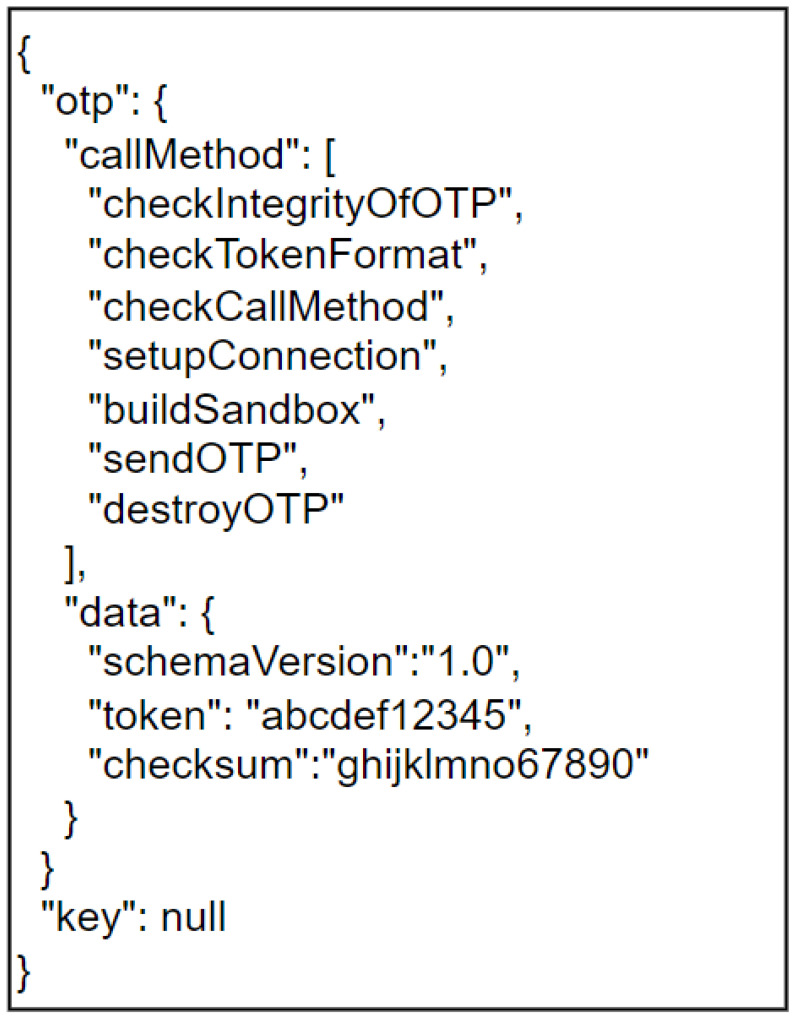
Clear version of the sandbox.

**Figure 4 sensors-23-01336-f004:**
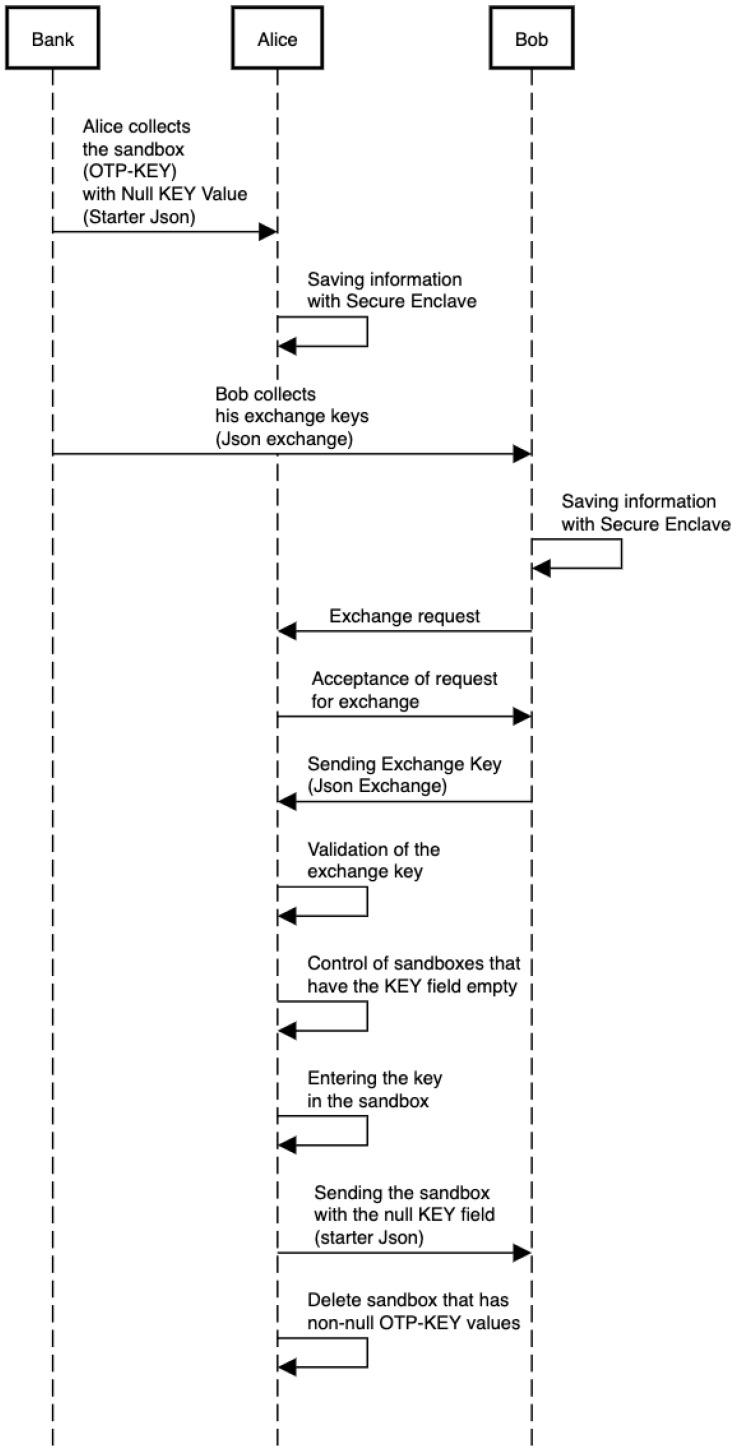
UML sequence diagram showing how the internal protocol works.

**Figure 5 sensors-23-01336-f005:**
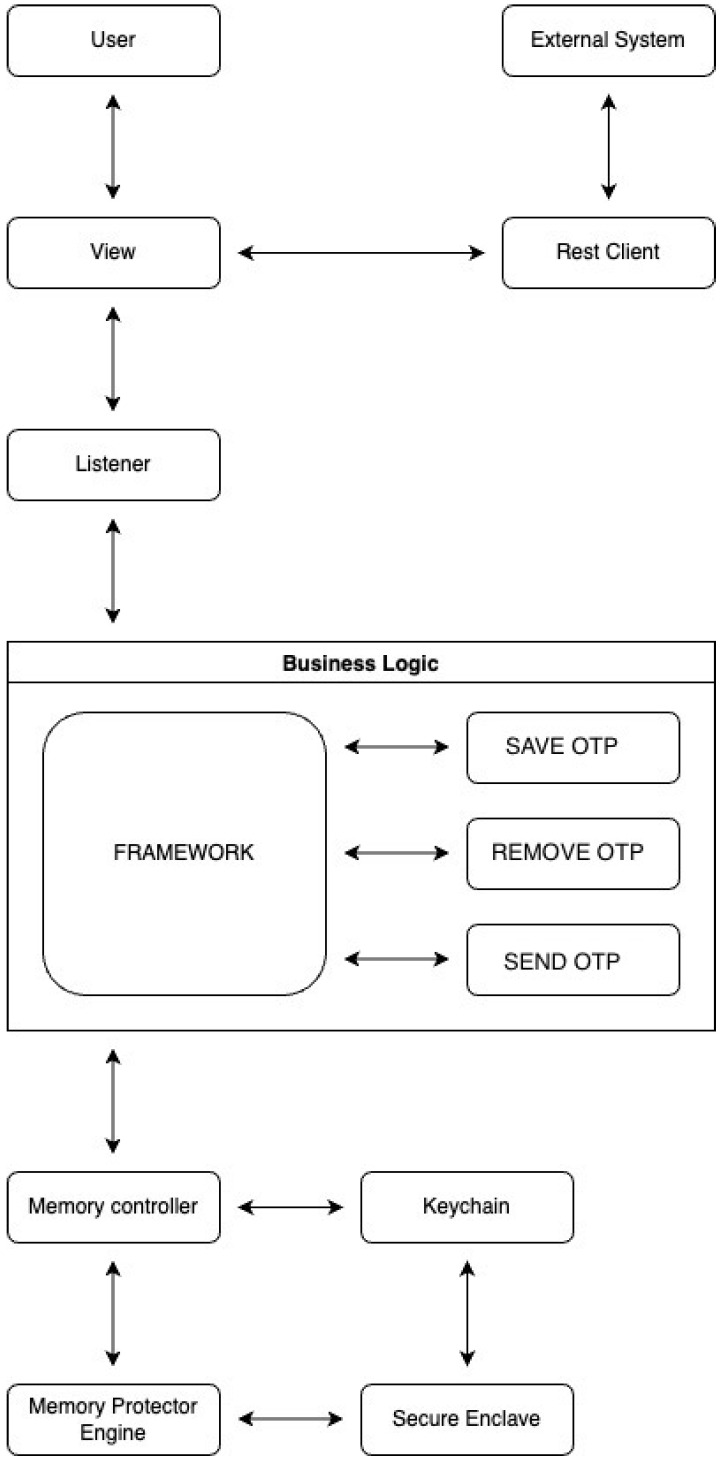
Logical structure of the application using a framework for isolated management of OTPs.

**Figure 6 sensors-23-01336-f006:**
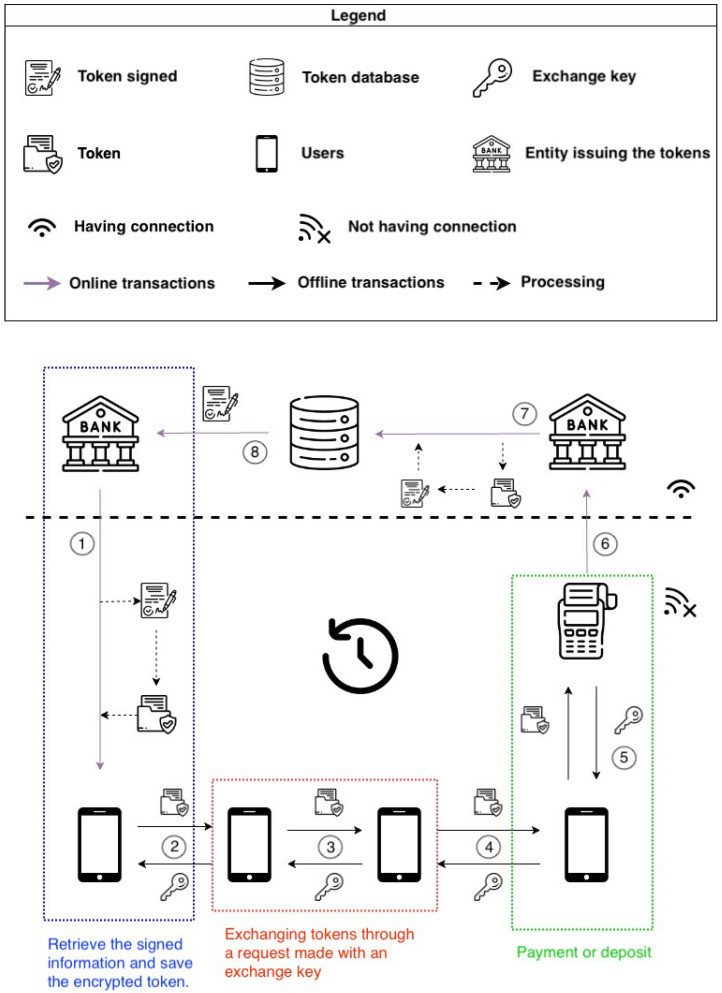
Life cycle of a transaction using an asymmetric key.

**Figure 7 sensors-23-01336-f007:**
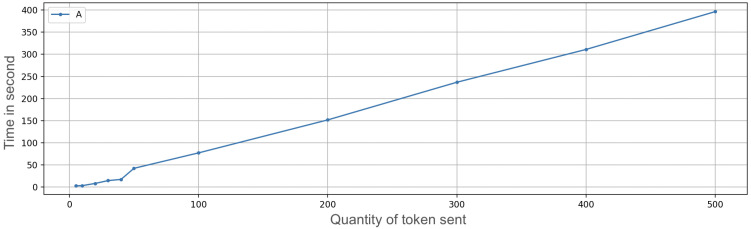
Exchange token time on iPhone 7 with secure enclave execution measuring average (A) of ramp test.

**Figure 8 sensors-23-01336-f008:**
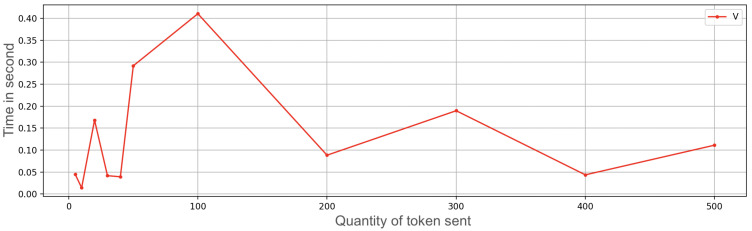
Exchange token time on iPhone 7 with secure enclave execution measuring variance (V) of the ramp test.

**Figure 9 sensors-23-01336-f009:**
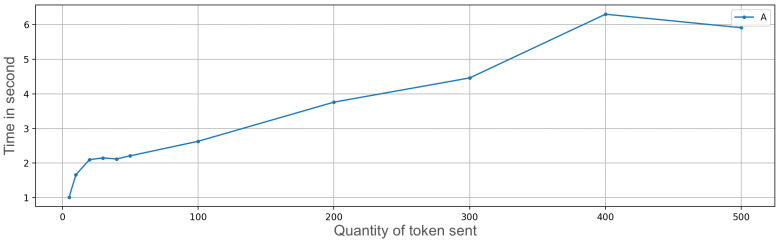
Exchange token time on iPhone 7 without the secure enclave execution measuring average (A) of the ramp test.

**Figure 10 sensors-23-01336-f010:**
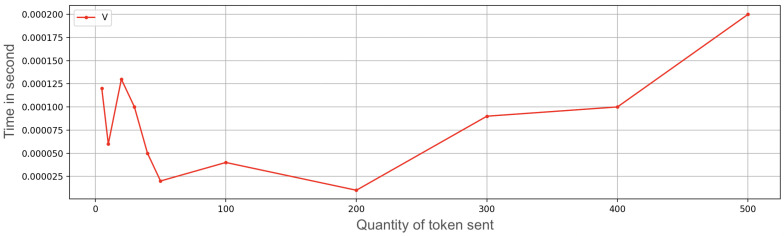
Exchange token time on iPhone 7 without the secure enclave execution measuring variance (V) of the ramp test.

**Figure 11 sensors-23-01336-f011:**
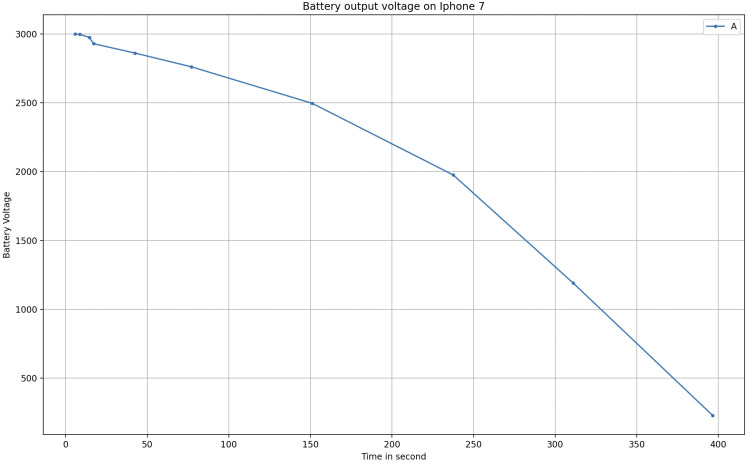
Battery level during the ramp test.

**Figure 12 sensors-23-01336-f012:**
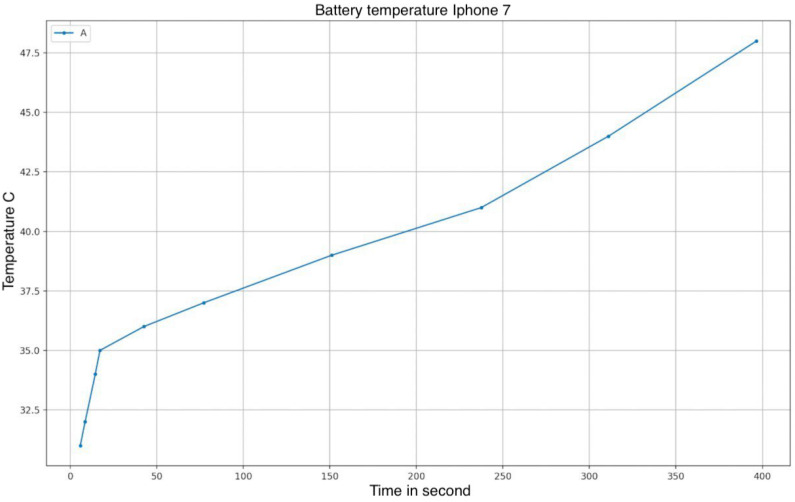
Battery temperature during tests.

**Figure 13 sensors-23-01336-f013:**
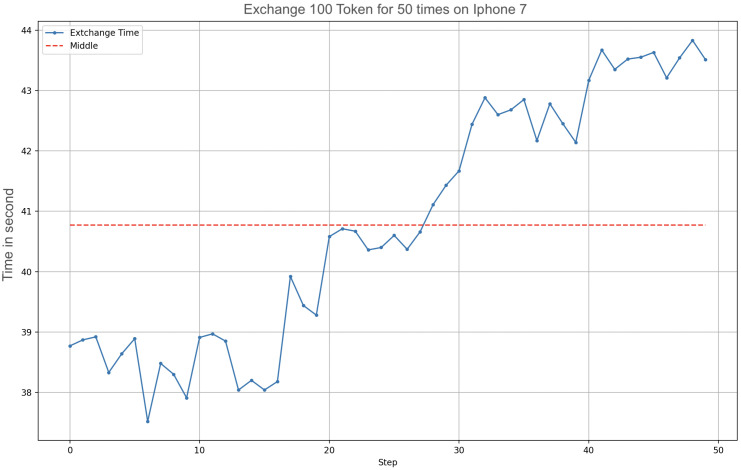
Sending 100 tokens, 50 times.

**Figure 14 sensors-23-01336-f014:**
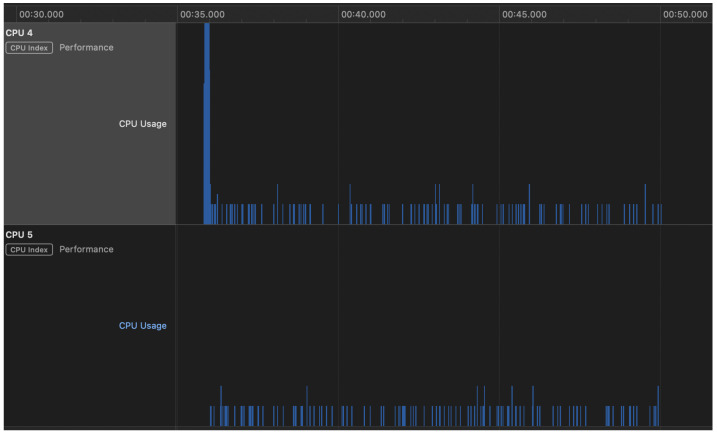
Percentage of CPU usage on the iPhone during a transfer of 100 OTPs.

**Figure 15 sensors-23-01336-f015:**
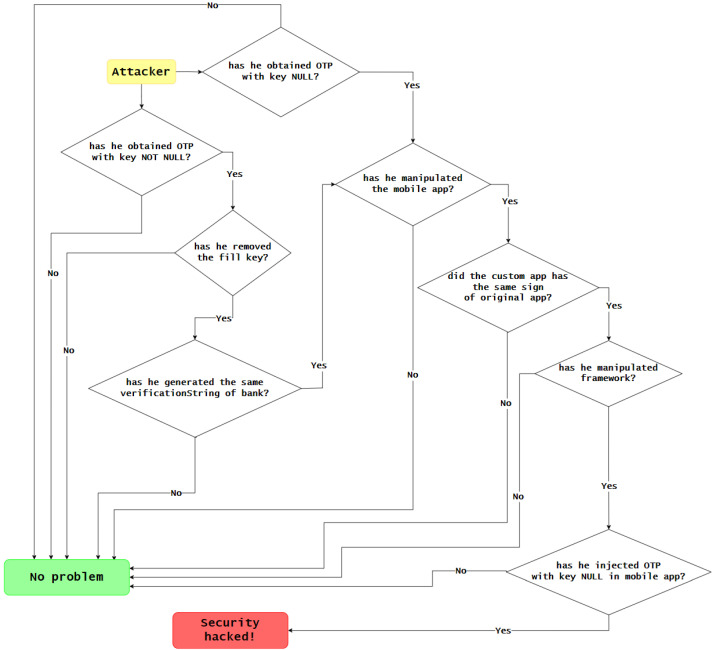
Steps an attacker must take to breach the system.

**Table 1 sensors-23-01336-t001:** Configurations where the prototype can transfer and receive tokens.

Wi-Fi	Bluetooth	Aeroplane	Working
	Low Energy	Mode	
ON	OFF	OFF	YES
OFF	ON	OFF	YES
ON	ON	OFF	YES
OFF	OFF	OFF	YES
OFF	OFF	ON	NO
ON	OFF	ON	YES
OFF	ON	ON	YES
ON	ON	ON	YES

**Table 2 sensors-23-01336-t002:** Security analysis.

Security Analysis	Critical Point	Covered
Vulnerability analysis	The bank generates OTPs with a key NULL that can be intercepted by a possible attacker.	Yes. The use of complex authentication-based security protocols allows the problem to be solved.
	The application could be hacked.	Yes. Even if the application were to be hacked, the core resides in the pre-compiled library.
	The framework could be hacked.	Yes. The pre-compiled library could be attacked in some way; therefore, continuous updates and penetration testing must be performed to ensure security but this aspect will not be covered in this paper.
	An attacker could go between two users who are exchanging e-coins and perform a MITM attack.	Yes. The exchange of OTPs is done over a secure channel by exploiting the Diffie–Hellman algorithm with zero knowledge proof, as presented in Section 2.
Input data analysis	An attacker can manipulate the OTP.	Yes. The model of the OTP contains a series of checks to verify the integrity of objects and the validity of the information. These checks are designed to ensure that the program is able to process data correctly and to prevent errors or vulnerabilities that could be exploited by an attacker.
Dependency analysis	An attacker can manipulate the OTP and SE.	Yes. Both can be safe if properly configured; therefore, there are no dependencies that expose the proposed solution to further vulnerabilities.
Authorization analysis	An attacker can manipulate the application and attack the bank	Yes. If an attacker manipulates the application, the integrity changes and all kinds of connections are rejected.
	An attacker can manipulate the application and attack SE	Yes. Secure memory can be accessed only with biometric information. If an attacker finds a phone with some OTP inside, he/she cannot extract or use it.

**Table 3 sensors-23-01336-t003:** Comparison table.

	Cash	Our Approach	[26]	[18]	[21]	[22]	[24]	[27]	[28]
Decentralized payments	Coin issuing is centralized but exchanges are decentralized	Coin issuing is centralized but exchanges are decentralized	Yes, but needs multiple nodes for validation	Yes, but needs multiple nodes for validation	Coin and profile issuing is centralized but exchanges are decentralized	Coin and profile issuing is centralized but exchanges are decentralized	No, there is an online check	Coin issuing is centralized but exchanges are decentralized	Coin issuing is centralized (trustees) but in offline mode, the payments are decentralized
Internet network use	No	No	Yes	Yes	No	No mention on the network interface used	Yes, there is an online database of the group manager	No	Have an offline variant
Information security and anonymity	Secure and anonymous	Secure and anonymous	The encryption process is entrusted to successive nodes, so confidentiality is not independent from the other involved devices	Some concerns about anonymity are arising	An account ID is needed to perform transactions	Banks and TTP can ask users to self-decrypt payment information	The group manager can trace the user; moreover, there is an online check	Fair-tracing: identity can be revealed in case of illegitimate usages	Requires user interaction with a trustee; moreover, user information is attached to coins in offline mode
Performance	No energy spent during transactions	Increased execution time and energy expenditure.	Generally less energy expenditure and reduced timelines.	Transaction executed in milliseconds to seconds. Energy absorbed is not specified	Time and energy consumption are not provided	Time and energy consumption are not provided	Time and energy consumption are not provided	Time and energy consumption are not provided	Time and energy consumption are not provided
Need of a TTP	No	Yes, only for signing OTPs	No	No	No	No	No	No	
Programmability of coins	No	Yes	No	No	No	No	No	No	No
Additional hardware	No	Trusted Execution Environment (available on modern CPUs)	No	No	Secure Element	No	No	No	No
Timeliness in double-spending prevention	N/A	Yes	No	No (based on asynchronous penalties)	Yes	No	Yes	When the double-spending coin is deposited, the bank can trace the double-spender	
Coins are ’markable’	No	Yes	No	No	No	No	Yes	No	No
Requires a bank account	No	No	No	No	No	Yes	Yes	Yes	
Payment fees	No	No	Yes	Yes	No	No	No	No	
Battery [mV]/Time [s]	N/A	Increases as the transaction increases. Few tokens exchanged: 2/15. Many tokens exchanged: 500/100	Moderately stable: 200/50 [min]	N/A	N/A	N/A	N/A	N/A	N/A
Battery [%]/Time [s]	N/A	Increases as the transaction increases. Few tokens exchanged: 1/15. Many tokens exchanged: 30/100	Moderately stable: 20/50 [min]	N/A	N/A	N/A	N/A	N/A	N/A
Temperature [C]/Time [s]	N/A	Increases as the transaction increases. Few tokens exchanged: 3/2. Many tokens exchanged: 3/100	Moderately stable: 6/10 [min]	N/A	N/A	N/A	N/A	N/A	N/A

## Data Availability

Not applicable.

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
