# Peer review of "A Sustainable Approach to Delivering Programmable Peer-to-Peer Offline Payments"

_sensors, 2023, doi:10.3390/s23031336_

Round 1

Reviewer 1 Report

In this paper, authors have presented a sustainable offline P2P payment scheme facing the double spending problem by means of a One Time Program (OTP) approach. The obtained results reveal that the proposed payment scheme is energetically sustainable. There are some following concerns related to the submitted manuscript:

# Authors have laid down a scenario in the introduction section regarding that emphasis on changing the use of traditional currency to digital currency. However, conducted experiments are more based on energy consumption rather than evaluating the proposed technique on other performance evaluation parameters. 

#Proposed technique/framework must be evaluated for its security, and efficiency concerning to no. of transactions per unit time handled at the server side.

#Authors have designed this technique for iOS. How the proposed technique will meet the challenges associated with e-SIM. As in e-SIM the concept based on OTP will not be effective and anybody can perform the transactions.

#Caption of Fig 5 need to be updated. 

#Fig 7 consist of two sub figures. All the subfigures must be identified with its captions. Also, what does legends A & V refers in this figure?

#Authors have considered the wrong parameters for performance evaluation. I don't think that anybody is free to perform 100+ transactions in a day so opted Battery temperature and Battery usage parameters are not correct. However, authors should adopt some cryptographic measures that evaluate the security of the proposed technique.

 #check for typos like guardanteed, etc..

Author Response

  1. Authors have laid down a scenario in the introduction section regarding that emphasis on changing the use of traditional currency to digital currency. However, conducted experiments are more based on energy consumption rather than evaluating the proposed technique on other performance evaluation parameters.

Thank you for pointing it out, yes we made an evaluation only proving the energetic sustainability of the proposed schema. Nevertheless, as per your suggestion, we decided to include a performance evaluation based on a basic metric like the CPU usage on the mobile devices involved in the transaction. Moreover, we made more explicit the time metric, that was already present.

  1. Proposed technique/framework must be evaluated for its security, and efficiency concerning to no. of transactions per unit time handled at the server side.

We understand the importance of the topic, hence we decided to involve another colleague in the manuscript who is currently working on the security assessment of the protocol. So we added a specific subsection (3.3) in the evaluation section were we report on the security evaluation. This forced us to expand the 2.7 subsection (“Proposed method”) to add some protocol details, involved in the security evaluation. Please notice that, being a P2P protocol schema, the server is not directly involved in the transaction, so we did not include performance metrics on the server side. Thanks.

  1. Authors have designed this technique for iOS. How the proposed technique will meet the challenges associated with e-SIM. As in e-SIM the concept based on OTP will not be effective and anybody can perform the transactions.

The concept is very interesting and was partially included in the comparison with the Secure Element alternative approach. Hence we added a specific paragraph ad the end of the discussion section to clarify how the proposed schema would deal with an e-Sim instead of a TEE. Many thanks for your suggestion!

  1. Caption of Fig 5 need to be updated. 

Done, thank you.

  1. Fig 7 consist of two sub figures. All the subfigures must be identified with its captions. Also, what does legends A & V refers in this figure?

Done, thank you. Now we have two pictures (fig. 7 and 8). We proactively made the same thing to another picture which needed to be split as well (fig. 9 and 10).

  1. Authors have considered the wrong parameters for performance evaluation. I don't think that anybody is free to perform 100+ transactions in a day so opted Battery temperature and Battery usage parameters are not correct. However, authors should adopt some cryptographic measures that evaluate the security of the proposed technique.

Thank you for your observation. Consider that at the current state of the protocol each transaction involves only 1 digital currency (e.g. 1$ or 1€), so it could be very easy for a payer to settle 100+ atomic transactions in a day. We better clarified this concept at the beginning of the evaluation section. Having this current limitation, in our opinion it does make sense to check how many transactions could be settled with a single battery full charge. Nevertheless, as already explained in point 2, we added a specific security evaluation section. Thank you!

  1. check for typos like guardanteed, etc..

We run a spell check and fixed lot of typos, many thanks!

Reviewer 2 Report

This paper we present a sustainable offline P2P payment scheme facing the double spending problem by means of a One Time Program (OTP) approach. The approach consists in wiping the business logic out of clients’ app and letting financial intermediaries inject certified payment code in the user device, that will execute asynchronously and offline at the time of payment. The idea seams interesting and novel. However, I have following suggestions to improve its overall quality.

 1.      The Introduction section needs improvement. The authors should add the following subsections: Main contributions.

2.      The authors should try to explain clearly the differences and connections between the proposed approach and blockchain-based digital currencies.

3.      In the do-offer method, if a transaction fails because of a power failure or other reasons, how can the integrity of this transaction be guaranteed?

4.      In the same field of interest there are some latest papers that would increase the technical strength of the article, Please analyze and cite the following articles, such as, - A Semi-centralized Trust Management Model Based on Blockchain for Data Exchange in IoT System. - A Blockchain-empowered Federated Learning in Healthcare-based Cyber Physical Systems. - VRepChain: A Decentralized and Privacy-preserving Reputation System for Social Internet of Vehicles Based on Blockchain.

5.      The authors should discuss how secure the system is if the Trusted Execution Environment is attacked.

6.      It needs to be argued, is there a possibility of man-in-the-middle attack?

Author Response

  1. The Introduction section needs improvement. The authors should add the following subsections: Main contributions.

Yes we agree with your concern, we added the section 1.1 “Main contribution”. Thanks!

  1. The authors should try to explain clearly the differences and connections between the proposed approach and blockchain-based digital currencies.

Thank you for pointing it out, it was needed a specific paragraph in the discussion section. We wrote it. Many thanks.

  1. In the do-offer method, if a transaction fails because of a power failure or other reasons, how can the integrity of this transaction be guaranteed?

We added some consideration about this possibility in the security evaluation subsection (3.3), a totally new part that we added because of other concerns. Thank you very much!

  1. In the same field of interest there are some latest papers that would increase the technical strength of the article, Please analyze and cite the following articles, such as, - A Semi-centralized Trust Management Model Based on Blockchain for Data Exchange in IoT System. - A Blockchain-empowered Federated Learning in Healthcare-based Cyber Physical Systems. - VRepChain: A Decentralized and Privacy-preserving Reputation System for Social Internet of Vehicles Based on Blockchain.

The papers are very interesting, they enrich our state of the art. We added it.

  1. The authors should discuss how secure the system is if the Trusted Execution Environment is attacked.

Thank you, we included the evaluation of this possibility in Table 2.

  1. It needs to be argued, is there a possibility of man-in-the-middle attack?

The answer is yes, so we added the subsection 2.6 where we explain the technique used in our protocol to avoid MITM.  Thank you!

Reviewer 3 Report

I have few comments:

1. The authors claim that the proposed method is novel in 4 segments; including double coin spending, privacy , logical coins and less power consumption. However, lets take each parameter of these 4 segments for example privacy - this term is not checked in later results. How do you ensure your algorithm ensure privacy. Did you generate certain cyber-attacks on your transactions process and guarantee security

2. Figure 1 is a standard available figure - it must be removed.

3. Figure 4 illustrating Alice and Bob (as 2 characters) is available in almost all books and power point lecture notes. Please do not reproduce standard text/figures again. These are not your contributions.

4.  In Figure 6, what is meant by key - is it symmetric or non-symmetric and is private or public?

5.  The ramp test (Figure 7) and its behavior needs more explanation of what is happening and why it is happening?

6. Figure 8 is straight forward figure - it would be interesting to see if the power consumption does not increase. What the authors want to prove in Figure 8 (just an increasing plot)

Author Response

  1. The authors claim that the proposed method is novel in 4 segments; including double coin spending, privacy , logical coins and less power consumption. However, lets take each parameter of these 4 segments for example privacy - this term is not checked in later results. How do you ensure your algorithm ensure privacy. Did you generate certain cyber-attacks on your transactions process and guarantee security

You are right, we did not discuss how our scheme ensures privacy. So we’ve added a specific paragraph at the end of section 4 about this topic. In general, we did not perform cyber-attacks simulations, nevertheless in the new version of the paper we involved another colleague who was working on the security of the protocol, and we added a new subsection called “Security evaluation” (3.3) where we perform a qualitative and systematic evaluation of the security of the protocol. We hope this is ok on your side. Thank you very much for your precious suggestion!

  1. Figure 1 is a standard available figure - it must be removed.

Yes, we added it for reader’s convenience. But actually it’s a standard figure available on the Web. We followed your suggestion and we removed it. Thank you!

  1. Figure 4 illustrating Alice and Bob (as 2 characters) is available in almost all books and power point lecture notes. Please do not reproduce standard text/figures again. These are not your contributions.

Thank you for pointing it out, nevertheless it is the UML sequence diagram of our protocol, we did not take the picture from other resources. We think that this picture is very important because it describes in a graphical way how the proposed schema works. If you kindly agree, we’d like to keep it. Many thanks.

  1. In Figure 6, what is meant by key - is it symmetric or non-symmetric and is private or public?

Big oversight! Our mistake, we forgot to specify it. Thank you very much for noticing it, we added a specific sentence in the text to specify that the key is asymmetric.

  1. The ramp test (Figure 7) and its behavior needs more explanation of what is happening and why it is happening?

The picture is quite straightforward (linear behavior), anyway we added a few words to describe what is happening during the test, including an estimation of the time needed to transfer a single OTP.

  1. Figure 8 is straight forward figure - it would be interesting to see if the power consumption does not increase. What the authors want to prove in Figure 8 (just an increasing plot)

Yes, the picture was quite straightforward, it was needed to compute an estimation of energy consumption per single OTP. But we added this information in the text and hence we removed Figure 8. This makes the paper more readable. Thank you!

Round 2

Reviewer 1 Report

It's an interesting work that involves lots of experiments and discussions.

Reviewer 2 Report

This is a revised version. All the comments that I gave last time have been well fixed, so I recommended this paper.

Reviewer 3 Report

The authors have addressed all of my previous concerns and have improved the paper a lot. If the paper is written using correct journal template, then the paper is acceptable